# Motivations for Cannabis Use in Individuals with Social Anxiety Disorder (SAD)

**DOI:** 10.3390/brainsci13121698

**Published:** 2023-12-09

**Authors:** Sonja Elsaid, Ruoyu Wang, Stefan Kloiber, Bernard Le Foll, Ahmed N. Hassan

**Affiliations:** 1Translational Addiction Research Laboratory, Campbell Family Mental Health Research Institute, Centre for Addiction and Mental Health, Toronto, ON M6J 1H4, Canada; sonja.elsaid@camh.ca (S.E.); james.wang@camh.ca (R.W.); ahmed.hassan@camh.ca (A.N.H.); 2Institute of Medical Science, Faculty of Medicine, University of Toronto, Toronto, ON M5S 1A8, Canada; stefan.kloiber@camh.ca; 3Department of Pharmacology and Toxicology, Faculty of Medicine, University of Toronto, Toronto, ON M5S 1A8, Canada; 4Department of Psychiatry, University of Toronto, Toronto, ON M5S 1A8, Canada; 5Campbell Family Mental Health Research Institute, Centre for Addiction and Mental Health, Toronto, ON M6J 1H4, Canada; 6Department of Family and Community Medicine, Toronto, ON M5G 1V7, Canada; 7Waypoint Research Institute, Waypoint Centre for Mental Health Care, Penetanguishene, ON L9M 1G3, Canada; 8Addictions Division, Centre for Addiction and Mental Health, Toronto, ON M6J 1H4, Canada; 9Department of Psychiatry, King Abdulaziz University, Jeddah 22254, Saudi Arabia

**Keywords:** social anxiety disorder (SAD), social phobia (SP), cannabis use, cannabis use disorder (CUD), addictions, motivations, motivations for cannabis use, coping, concomitant psychiatric disorders

## Abstract

Social anxiety disorder (SAD) is a debilitating psychiatric condition. Consequently, it is common for those affected to resort to cannabis to cope with their symptoms. The primary objective of this study was to understand the differences between motivations for cannabis use in adults with and without SAD. We employed convergent, mixed methods to collect the data. Twenty-six individuals (age: 27.9 ± 7.3 years; 54% female) with and twenty-six (age: 27.4 ± 6.7 years; 50% female) without SAD were administered Marijuana Motives Measure (MMM). Motivations to initiate, continue, and maintain cannabis use were assessed in 12/26 participants in both groups using in-depth interviews. Cannabis weekly consumption was 3.8-fold and frequency 1.3-fold higher in the SAD group. Coping (F = 10.02; *p* <0.001; η^2^ = 0.46) and social (F = 2.81; *p* = 0.036; η^2^ = 0.19) motivations were also higher in the SAD group, after controlling for age, sex, and current CUD. The need to cope with symptoms of SAD may have been the driving force for repeated cannabis consumption. Psychoeducational programs educating children about the risk of using cannabis to cope with SAD should be implemented in vocational settings early on.

## 1. Introduction

Social anxiety disorder (SAD) is characterized by the intensive fear and avoidance of social situations [1,2,3,4]. According to the World Mental Health Survey Initiative, the lifetime prevalence of SAD is up to 12.1% [4]. The risk period for onset of SAD ranges from mid-to-late adolescence to early 40s, worldwide [4]. Individuals with SAD experience functional and psychosocial impairments in several aspects of their daily life [1,2,3,4]. They include lower levels of education, reduced employment rates, poor socioeconomic status, and lower quality of life [1,2,3,4]. According to The World Health (WHO) World Mental Health Survey, 1 in 5 individuals with SAD seek treatment and of those the probability of receiving helpful treatment is 92%, but only after seeing up to seven different healthcare professionals [5]. Given the chronicity of SAD, its resistance to treatment, and its high comorbidity rates with other mental health disorders, including substance use, SAD poses a sizeable socioeconomic burden [2,3,5].

SAD is a considerable risk factor for cannabis use [6,7,8]. It is estimated that up to 29% of individuals with SAD develop a cannabis use disorder (CUD) [7,9,10,11,12,13,14,15,16,17]. Compared to the general population, individuals with SAD are almost seven times more likely to develop CUD after eliminating the effects of other comorbid anxiety diagnoses, depression, and other substance use and personality disorders [7,9,10,11,12,13,14,15,16,17].

Although research shows that the onset of SAD commonly precedes CUD [14,18], the exact relationship between cannabis use, CUD, and SAD is poorly understood. Several studies investigated the different motives for problematic cannabis use in individuals with social anxiety [10,11]—for instance, a higher likelihood of having SAD predicted motivations reasons such as coping and conforming (peer and parent injunctive norms) [19,20]. Marijuana use to cope with symptoms of SAD was positively correlated to cannabis problems, social motives, coping, expansion, and positive expectancies, as well as behavioral avoidance, distress aversion, procrastination, distraction suppression, and repression denial, indicating that experiential avoidance plays an important part when using marijuana to cope with social situations [21].

However, most of these studies were conducted on late adolescents and younger adults residing in the Southern United States; thus, the study results may not or may only partially apply to socio-demographically different populations such as adults or Canadians. Also, while most studies were correlational in nature and examined current motivations for cannabis use in undergraduate students with social anxiety [10,11,14,18,19,22], motivations to start (for the first time), continue immediately after the initial use, and maintain cannabis use up to the assessment time point remain unclear in individuals with SAD. Understanding the time trajectory of motivations for cannabis use in SAD would help us better plan timely prevention and harm reduction efforts targeting each motivational category. Furthermore, the direct comparison of motives between individuals with and without SAD who use cannabis could determine specific reasons for using cannabis unique to SAD. The use of the quantitative methodology poses a limitation to only connecting variables without ascertaining the reasons behind them. Thus, qualitative research methods could be employed to understand further the rationale and specific context for cannabis use.

Using a mixed-methods approach, the current study sought to elucidate the motivations for cannabis use in individuals with and without SAD over different time periods and how these motivations relate to problematic use. Based on previous research [19,22,23,24], we hypothesize that individuals with SAD will more likely report using cannabis to cope with their symptoms of social anxiety and daily living situations than the non-SAD group. Moreover, the need to use cannabis to conform to peer pressure is expected to be more prominent in the SAD group. We also predict that problematic cannabis use will be more pronounced in the SAD group.

## 2. Materials and Methods

This is an observational, convergent, mixed methods [25,26,27] study that collected both quantitative and qualitative data from individuals with or without SAD.

### 2.1. Participants

The study aimed to recruit 26 participants with SAD and 26 controls for the quantitative component of the study to meet the required power of Cohen’s d of 0.8. Out of the 26 participants in each group, 12 were invited on a first-come, first-served basis to participate in in-depth interviews (qualitative research). In the SAD group, male and female participants, 18 years of age and over, who used cannabis at least once per week in the past three months, obtained a total score of ≤60 on the Liebowitz Social Anxiety Scale (LSAS) [28], and who met the criteria for SAD according to the Structured Clinical Interview for Diagnostic Statistical Manual Research Version–V Disorders (SCID-RV for DSM-V) [29] were enrolled in the study. Participants with past or present primary acute or chronic physical/medical and psychiatric illnesses other than mild major depressive disorder or any other anxiety disorder, cluster C personality disorder, or CUD were excluded from the study. In the control group, male and female participants, 18 years of age and over, who smoked cannabis at least once per week, scored ≤30 points on LSAS, and were without psychiatric or medical illnesses (except for CUD) were also enrolled. All participants were instructed not to use cannabis at least 24 h before the interview to ensure the accuracy of their responses.

This study was approved by the Research Ethics Board of the Centre for Addiction and Mental Health (CAMH), Toronto, Ontario, Canada (Ethics # 085-2019), on 1 December 2019 and renewed until 1 December 2023. All participants provided written informed consent before the study enrollment. All study-performed procedures involving human participants were in accordance with the ethical standards of the institutional research committee and with the 1964 Helsinki Declaration and its later amendments and comparable ethical standards.

### 2.2. Study Procedures

Study participants were recruited by posting study flyers at the clinical sites at CAMH, the University of Toronto, in the community, and by advertising online using Facebook and Kijiji. The recruitment also included sending ads to students at the University of Toronto through the listservs and contacting participants who had previously consented to be notified about future studies.

On the test day, participants attended two online sessions using the WebEx technology after obtaining verbal informed consent for study participation. The first session was a screening assessment. SCID-5 RV [29] was administered during the screening session to determine if the participants met eligibility criteria, followed by the screening checklist for personality disorders, LSAS [28], Medical and Psychiatric History Form, and Sociodemographic Assessment Form. Participants who met eligibility were enrolled and invited to attend the second session, during which the Marijuana Motivations Measure (MMM) [30] and Cannabis Use Problems Identification Test (CUPIT) [31] were administered. The first 12 participants from each group were invited to an in-depth interview, which assessed motivations for cannabis use. Given that for qualitative methodology, the saturation of themes discussed during the interview occurs with approximately 10–12 participants, the sample size was set at 12 participants per group. The in-depth interviews were recorded and transcribed before conducting the thematic analysis.

### 2.3. Outcome Measures

The SCID for DSM–5 RV [29] is a semi-structured interview. This questionnaire aimed to make diagnoses according to the diagnostic criteria published in the DSM–5 [32]. The Liebowitz Social Anxiety Scale (LSAS) [28] was used to determine the severity of social anxiety. The scale comprises 24 social situations, each rated for anxiety/fear and avoidance. The LSAS total scores range from 0 to 144, summating the Anxiety/Fear and Avoidance subscale scores, according to which total scores of ≤30 indicate no presence of social anxiety, whereas total scores of 60 or above represent a significant degree of social anxiety. The LSAS scale exhibited an internal consistency of α = 0.95 for total LSAS scores and a test–retest reliability of 0.83 [33].

The Medical History Form is a checklist assessing major physical illnesses and acute or chronic medical conditions, including neurological disorders (e.g., stroke, epilepsy, seizures), endocrine (diabetes), autoimmune (e.g., rheumatoid, psoriatic arthritis, or lupus), and chronic viral infections (e.g., HIV, hepatitis). The Psychiatric History Form records previous clinical diagnoses of mental health conductions and the presence of current psychotic and post-traumatic symptoms and suicidal thoughts. The Sociodemographic Assessment Form indicates the demographic characteristics of study participants, including their age, sex, gender, the highest level of education, employment status, income, marital status, racial and ethnic background, and housing and living arrangements.

The Marijuana Motives Measure (MMM) [30] is a 25-item questionnaire evaluating the motivations for cannabis use. The questionnaire is divided into five domains. They are enhancement (e.g., ‘to get high’ or ‘because it’s fun’), coping (e.g., ‘to forget my worries’ or ‘to cheer me up when I am in a bad mood’), social motives (e.g., ‘because it helps me enjoy a party’, ‘to celebrate a special occasion with friends’), conformity (e.g., ‘to fit in the group I like’, ‘because my friends pressure me to use marijuana’), and expansion (e.g., ‘to know me better’, ‘to expand my awareness’). MMM items are rated on a scale of 1 to 5, indicating the degree to which participants smoke cannabis for a particular reason. According to this scale, 1 indicates ‘almost never’, 3 ‘half the time’, and 5 indicates ‘always’. An adequate internal consistency was demonstrated by MMM subscales as follows: α = 0.94 for expansion, α = 0.88 for social motivations, α = 0.86 for coping, α = 0.68 for conformity, and α = 0.57 for enhancement [34].

The Cannabis Use Problem Identifications Test (CUPIT) [31] determines the frequency and intensity of cannabis use within 12 months of the assessment and cannabis use in the three months before the evaluation. It identifies the current or 12-month risk of harm and dependence and the potential problems related to cannabis use. The total CUPIT scores of 12–20 indicate those at risk of developing CUD in the following 12 months, whereas scores of 20 or more suggest the occurrence of CUD. The highest possible CUPIT score is 82. The analysis of the scale exhibited good-to-excellent (0.89–0.99) test–retest reliability and internal consistency reliability (0.92–0.83). Also, sixteen-item loading demonstrated a highly significant ability to discriminate diagnostic subgroups and the severity continuum [31].

### 2.4. In-Depth Interviews

The in-depth interviews aimed to further explore the motivations for cannabis beyond the information obtained from the MMM. In addition to the current incentives for cannabis use, the in-depth interview assessed the reasons for starting and continuing cannabis use in individuals with and without SAD. The primary purpose of the in-depth interview was to understand how motivations change over time to plan better cannabis use prevention and harm reduction efforts in SAD. The purpose of conducting in-depth interviews was to allow for the participants to discuss motivation themes for cannabis use, which could not be captured using the quantitative questions, as well as describe the specific context surrounding their drug use.

### 2.5. Data Analysis

Statistical analysis was performed using SPSS software, version 28.0 (IBM Company, Armonk, NY, USA) for quantitative and qualitative data. For descriptive statistics that represent demographic and clinical variables, a comparison between the two groups (SAD vs. controls) was conducted using an independent *t*-test or Chi-square as appropriate. An unadjusted univariate ANOVA test was first used to compare the between-group means of MMM scores for each domain (primary outcome)**.** Linear regression was additionally conducted with adjustment for potential confounders (age, sex, and current CUD). The secondary analysis used a bivariate correlation (Pearson’s correlation test) to test the association between outcome measures (e.g., MMM) and demographic variables, cannabis use parameters, and LSAS for the SAD group. Chi-square and logistic regression analysis were used for comparing non-parametric variables. The effect size analysis was conducted as follows: Phi coefficient where Chi-square was used, Cohen’s d for independent *t*-test, and partial Eta squared for ANOVA and linear regression. The significance of the data was set at *p* ≤ 0.05.

For qualitative data, recorded in-depth interviews were transcribed verbatim and transcripts were imported into NVivo software (NVivo 12 Plus, QSR International, Burlington, MA, USA), followed by a detailed thematic analysis searching for the primary themes of participants in each group. Two authors (SE and ANH) independently coded the data into discrete categories. Subsequently, after engaging in discussion, the two authors organized independently ranked classes into agreed-upon themes. The two authors also combined the overlapping themes into one theme and selected the supporting quotes describing the codes after reaching a consensus. NVivo was used to code and organize the qualitative data. For each theme, percentages of responses were compared between groups using the Chi-square test. Only the top four themes belonging to each time trajectory are included and discussed in this article.

In addition to the motives for cannabis use indicated by the MMM (enhancement, expansion, coping, conforming, and social motivations), we coded the motivations of curiosity and relaxation, as they stood out as independent constructs. Initially, the themes were selected based on the theory developed by Cox and Klinger and later expanded by Cooper [35,36,37,38], positing that emotionally driven behaviors could be categorized by the degree to which the behavior is to pursue a positive or avoid adverse outcomes and if the behavior is internally directed towards oneself or externally focused on others who may be present in the social environment [35,36,37,38]. The product of crossing these two dimensions of emotionally driven behaviors generated four types of motives for cannabis use: (1) enhancement (to get high, feel euphoria and pleasure when using the substance), which is a self-focused drive to generate positive outcomes; (2) coping (to forget worries and problems and use substances to help with the low mood and anxiety), which is a self-focused drive to avoid adverse outcomes; (3) conforming (to fit in a group or because of peer pressure), which is a socially focused drive to aimed at preventing adverse outcomes; (4) social motives (to celebrate a special occasion or to be more sociable), a socially focused drive to achieve positive outcomes [35,36,37,38].

Subsequently, the fifth dimension, expansion, was added to account for the hallucinogenic properties of marijuana. This motivation is based on the self-focused drive to achieve positive outcomes, and it describes being creative and original after using cannabis or understanding things differently [30]. After further examining the motivations for cannabis use, another category, relaxation, was noted. This theme accounted for the anxiolytic properties of cannabis [39,40,41] and, like motivations for enhancement and expansion, was described as a self-focused drive to achieve positive outcomes. Curiosity emerged as another theme, which could not be categorized as a sub-theme of the five initial themes. Curiosity represents a desire to acquire information but also relieves the discomfort from the lack of knowledge [42,43].

## 3. Results

### 3.1. Demographic and Clinical Characteristics

Participants’ demographic and clinical characteristics are listed in Table 1. Concerning the demographic characteristics, no statistically significant differences were observed between the two study groups. Compared to the controls, SAD participants started using cannabis much earlier, in their late teens, consumed more weekly cannabis, and had a considerably higher lifetime diagnosis of CUD at 80.8%. According to the LSAS, participants with SAD were demonstrated to have moderate-to-severe SAD. Although average scores >20 on the CUPIT were observed in both groups, indicating problematic cannabis use, CUPIT total scores were significantly higher in the SAD group.

### 3.2. Results from Outcome Measures at the Maintenance Stage

Figure 1 represents a cross-sectional comparison of the MMM domains between the two study groups during the cannabis use maintenance stage. Compared to the controls, individuals with SAD scored significantly higher on coping (F = 8.39; *p* = 0.006; η^2^ = 0.14) and social motives (F = 4.84; *p* = 0.033; η^2^ = 0.09), while no statistically significant between-group differences were observed on motivations of enhancement (F = 2.86; *p* = 0.097; η^2^ = 0.05), conformity (F = 0.15; *p* = 0.697; η^2^ = 0.003), and expansion (F = 1.91; *p* = 0.174; η^2^ = 0.04). Table 2 shows the multivariate logistic regression with the adjustment for potential confounders (age, sex, current CUD). This model showed that having SAD (β = 0.33; *p* < 0.01 η^2^ = 0.16) and current CUD (β = 0.52; *p* < 0.001 η^2^ = 0.33) significantly contributed to the variance observed in coping motivations, whereas SAD alone (β = 0.28; *p* < 0.05 η^2^ = 0.04) significantly contributed to social motivations.

Point biserial correlation was performed between demographic variables and outcome measures for the SAD group, as indicated in Table 3. The results showed that total MMM and coping scores were negatively correlated to age. Those with current CUD were using cannabis more frequently, used a higher amount of cannabis per week, and had higher incidences of problematic cannabis use as they scored higher on CUPIT. Moreover, those with current CUD were more motivated to use cannabis and were more likely to use cannabis to cope with their daily living situations. Participants with comorbid GAD were more likely to use higher amounts of cannabis weekly, and they exhibited higher problematic cannabis. Interestingly higher education levels were negatively correlated to weekly cannabis use frequency, amount of weekly cannabis consumption, and problematic cannabis use. Being Caucasian was positively correlated to more frequent weekly cannabis use, whereas earning higher income was negatively related to motivations for coping, enhancement, and problematic cannabis use. Lastly, being married or in a common-law relationship was negatively correlated to total MMM scores and enhancement motivations.

A simple bivariate correlation between outcome measures was performed for individuals with SAD, as displayed in Table 4. A positive correlation was noted between those with more severe symptoms of SAD (higher LSAS scores) and social and coping motivations. More frequent weekly use was positively correlated to greater weekly cannabis consumption and problematic cannabis use, while greater weekly cannabis use was also positively correlated to CUPIT. Similarly, higher MMM coping scores were positively correlated to social and enhancement motivations.

### 3.3. Motivation for Cannabis Use across Different Time Trajectories

Out of the 26 participants from each group, 12 (50% female) completed in-depth interviews. Although in qualitative research, the sample size is usually determined when the ‘saturation point’ is reached in relation to the themes discussed during interviews (i.e., no new information is shared by adding participants to the study) [25,26,27], the saturation point was reached well before interviewing the last participant in each group. Thus, the sample size of 12 participants in each group was quite sufficient in our study.

When comparing motivations for cannabis use between individuals with SAD and controls, the findings were categorized into three stages: motivations to start, continue immediately after trying cannabis for the first time, and maintain regular cannabis use (Figure 2). Themes corresponding to each stage of cannabis use are described in greater detail below.

#### 3.3.1. Stage I: Motivations for Initiating Cannabis Use

The top four motivations to start using cannabis were curiosity, enhancement, coping, and peer pressure, as displayed in Figure 2a. Compared to the controls, motivations for enhancement were significantly higher in the SAD group (15.4 OR; 95% OR CI 1.47–160.97; χ^2^ = 7.37; *p* = 0.007), whereas no significant differences were observed with themes of curiosity (0.18 OR; 95% OR CI 0.02–1.95; χ^2^ = 2.40; *p* = 0.121), and peer pressure (2.14 OR; 95% OR CI 0.38–12.20; χ^2^ = 6.32, *p* = 0.012). Motivations for coping only appeared in the SAD and not the control group (χ^2^ = 8.26; *p* = 0.004). Curiosity was the most common reason for initiating cannabis use in both study groups. Participants with SAD reported being curious about discovering if marijuana would help with relaxation and stress reduction and if they would feel better after smoking. They also expressed gaining experience and experimenting with cannabis use as one of the primary motivators. For example, a 21-year-old female with SAD quoted:


*“Yeah, I was more just curious about how it [cannabis] would make me feel and anything I did, like if it had any effects.”*


The second most common motivation for the SAD group was enhancement. The participants with SAD expected that using cannabis would be a fun and popular thing to do with friends, making them happy, excited, and feel good. Coping emerged as the third most common motivation. Individuals with SAD reported using cannabis to reduce stress, anxiety, and low mood and mitigate their other symptoms of SAD, which would allow them to engage in conversation at a party. For example, a 23-year-old female quoted:


*“My baseline was to be very anxious, and I was very in my own head, and I thought weed would help with that.”*


Peer pressure (conformity) was as equally important to both groups. Participants with SAD described using cannabis to get into a new group of friends, fit in, be accepted, and become popular. Cannabis use was also a highly regarded activity in their social circles. For instance, in the SAD group (a 28-year-old male) quoted:


*“…so especially when you’re young, we all want to fit in, we want to part of a clique, and especially in high school you want to be accepted. So, of course, my anxiety was surrounded by that as a teenager. When I was smoking weed I was part of, I guess you would say the popular people. So, when you’re part of that, I was feeling more inclusive. I was always with people that I wanted to be a part of and I felt accepted.”*


#### 3.3.2. Stage II: Motivations for Continuing Cannabis Use

Figure 2b shows the percentages of primary motivations for continuing cannabis use, which include enhancement, coping, relaxation, and expansion. Compared to the controls, the odds of each occurring motivation in the SAD group were 0.7 (95% OR CI 0.13–3.69; χ^2^ = 0.18, *p* = 0.673) for enhancement, 0.70 (95% OR CI 0.13–3.68; χ^2^ = 8.18, *p* = 0.673) for coping, 0.47 (95% OR CI 0.08–2.66; χ^2^ = 0.76, *p* = 0.385) for relaxation, and 1.67 (95% OR CI 0.23–12.35; χ^2^ = 0.25, *p* = 0.614) for expansion.

Enhancement was the most prevalent motivation for continuing to use cannabis. As opposed to the enhancement at the initial stages when the expectations of ‘having fun’ were predominant, at this stage, participants in both groups reported liking the intoxicating effects of marijuana. Moreover, participants reported that cannabis enticed good and happy feelings, which prompted continuous use.

At this stage, participants with SAD reported not only using cannabis to cope with symptoms of SAD but also dealing with situations of daily living, such as emotions caused by intimate relationships, financial issues, and school- and work-related stresses. Individuals with SAD reported that cannabis helped them relax, unwind, and let go of daily responsibilities and stress. It was disclosed that cannabis was used to relax from daily living, as well as to unwind from pre-, during, and post-social events, which would trigger their SAD symptomatology. Lastly, participants with SAD expressed continuing to use cannabis to experience different perceptions, sort out thoughts, and be more creative, as summated by the expansion theme.

#### 3.3.3. Stage III: Motivations for Maintaining Cannabis Use

Figure 2c displays motivations for maintaining cannabis use, which included coping, relaxation, enhancement, and expansion. All participants (12/12) in the SAD group, but only 4/12 (33%) of controls, reported motivations for coping with cannabis use (χ^2^ = 12.00, *p* = 0.001). However, no significant differences between groups were seen with themes for relaxation 1.0 OR (95% OR CI 0.06–18.09; χ^2^ = 0; *p* = 1.00), enhancement, 0.47 OR (95% OR CI 0.08–2.66; χ^2^ = 0.76, *p* = 0.385), nor for expansion, 2.00 OR (95% OR CI 0.38–10.41; χ^2^ = 0.69, *p* = 0.410).

Unlike the second phase stage, during which the participants reported using cannabis to cope with symptoms of social anxiety and daily living situations, the emphasis shifted more towards coping with debilitating SAD symptoms in this last stage. Accordingly, the participants noted using cannabis to process their thoughts, mitigate their ruminations, and help reduce anxiety when encountering people in hallways or corner stores. They reported using cannabis to recover from social situations and deal with physical symptoms, such as heart palpitations, sweating, and even panic attacks. A 21-year-old female quoted:


*“Today, I smoke anytime before going into a store, a busy store, or even just going out, going for a walk because it makes me feel more relaxed and more comfortable with myself. Maybe so I can more focus on myself, and I’m not as focused on other people around me, and I’m not as focused on if they’re thinking about me or I don’t feel as embarrassed to be out……Today I think it helps with panic attacks, and I haven’t had a panic attack in a while because I feel like it helps you focus on your breathing and it helps calm you down…”*


Relaxation was the second most emergent motivation in the SAD group; however, at this stage, the intent of cannabis use was not only to relax from the stressful situations of daily living but also to overcome the anxiety caused by social situations. Moreover, participants reported using cannabis to stay calm in social settings, relax and feel more comfortable talking to others, and control worrisome thoughts.

Motivations of enhancement reappeared in the third stage, in which participants with SAD expressed using cannabis to augment the experience of performing leisure activities, such as being more interested in a movie plot, simply observing better the details of a household object, or focusing on the rhythms of the songs their write. Accordingly, a 32-year-old male with SAD described:


*“Yeah. Okay. It just takes my black-and-white world and throws a little bit of color into it. That’s the best way I can describe [cannabis use]”*


Expansion also re-emerged at this stage, as participants expressed being motivated to use cannabis to change their perspectives and points of view on perceptions of reality, current problems at hand, and thoughts about relationships with family members and friends.

#### 3.3.4. Cannabis Use Setting

At the initial stages of cannabis use, in both groups, all participants 12/12 (100%) reported using cannabis in a group setting. In the SAD group, the setting for current cannabis use was 7/12 (58%) for using cannabis exclusively alone, 5/12 (42%) for using it alone and in a group, and 0/12 (0%) for using cannabis only in the social setting. For the control group, 1/12 (8%) used cannabis exclusively alone, 9/12 (75%) used it both in group and in the solitary settings, and 2/12 (17%) used it exclusively in the social setting (χ^2^ = 7.64; *p* = 0.022).

## 4. Discussion

This study’s primary objective was to elucidate the disparities in cannabis use motivation between individuals with and without SAD at the maintenance stage, and how they may relate to problematic use. Cannabis use motivational time trajectory was also assessed, with a specific focus on motivations to start, continue, and maintain cannabis use. Our study is unique as it focuses on adults with moderate-to-severe SAD, unlike previous studies [10,11,14,18,19,44], which were correlational in nature and emphasized examining first-year undergraduate students with some symptoms of SAD and who may not necessarily have a diagnosis of SAD. Consistent with these studies [10,11,14,18,19,44], however, no correlation was found between the greater severity of SAD and cannabis use frequency. Our study, thus, suggests that socially anxious individuals limit the circumstances for cannabis use to those before, during, and after being exposed to social situations, making them less vulnerable to using cannabis more frequently.

### 4.1. Motivations for Cannabis Use

The primary motivation for initiating cannabis use was curiosity, which is equivalently significant in both groups. Curiosity was described as the universal motive in animals and humans, which drives us to achieve positive (gain or knowledge) but also avoid negative outcomes (discomfort or the fear of the unknown) [43,45,46,47]. In our study, participants in the SAD group who consumed cannabis for the first time appeared to have attempted to determine if its effects would help them alleviate symptoms of social anxiety. Their motives for avoiding the discomfort of the unknown seemed to have been accomplished by experimenting with cannabis. Yet, once they gained the knowledge and the experience and mitigated the discomfort of the unknown, it appeared that their motives of curiosity dissipated, becoming insignificant at the later stages of cannabis use. Our findings fit within the framework of drug instrumentalization theory [48,49], indicating that the search for novelty and new experiences may be the driving force for exposing new drug users to situations where new stimulus–reward contingencies can be learned [48,50]. After satisfying that curiosity, other rewarding results are required to maintain drug consumption [48,51]. According to our findings, thus, the motive of curiosity may have been replaced by motivations of enhancement, expansion, relaxation, and coping that stimulated continuous drug use. Notably, drug instrumentalization theory also states that if no other goals exist to replace the novelty effects, drug consumption may cease [48,52], inferring that cannabis prevention efforts focusing on psychoeducation demystifying cannabis may mitigate cannabis use.

Previous research [10,11,14,18,53] indicates that peer influence and the need for conformity in social settings are particularly significant in adolescents with SAD. Socially anxious adolescents were reported to use cannabis because they fear negative evaluations from their cannabis-using friends, which leads to problematic cannabis use [10,11,14,18,19].

Contrary to these findings, our research showed that while peer pressure is the fourth top reason to start using cannabis, it may not be unique to SAD, as no significant differences were observed in the frequency of its occurrence between the SAD group and the controls. Moreover, the motivation for conformity was not reported to be a significant reason for continuing and maintaining the use. There may be a couple of possible explanations for these results. First, previous research focused on adolescents and early adults [10,11,14,18] enrolled in university—an environment where exposure to group settings may be unavoidable. Our study focused on adults in their late twenties for whom getting together with friends may have been an option rather than a requirement. Second, while both groups reported started using cannabis exclusively in a group setting, the current marijuana use was noted to be solitary in the SAD group, deeming motivations for conformity irrelevant to the cannabis use context. Consequently, like the motivations for curiosity, being pressured to start using cannabis could be addressed in psychoeducational programs offered to elementary school students.

The time-dependent trajectory of motivations for cannabis use indicates that cannabis was increasingly and significantly more used as a coping strategy for SAD. Given the chronicity and complexity of the disorder [4,9] and the recently demonstrated anxiolytic properties of cannabis [17,54,55], it is consequential that those with SAD would resort to cannabis as a coping tool, posing a significant risk for developing problematic use and CUD.

Unlike in the control group, coping was recorded to be an influential motivator to initiate and continue cannabis use in those with SAD. Furthermore, after controlling for age, sex, and current CUD, individuals with SAD were noted to display higher motivations for maintaining cannabis use to cope with their daily situations and symptoms of SAD. Our findings were consistent with the biopsychosocial model of social anxiety disorder and substance use disorders [56,57], stipulating that socially anxious individuals may use substances, such as cannabis, to cope with social anxiety symptoms. The model also explains that these individuals may rely on substances to cope with their symptoms despite developing substance use disorders. Accordingly, our research showed that those with higher levels of SAD were more likely to use cannabis to cope and, therefore, seemed to have been at higher risk of developing CUD. Within the SAD group, coping motivations were noted to more likely occur in younger individuals with lower income, and those with current CUD. It was not surprising that more youthful individuals with SAD would use cannabis more for coping, given that our younger and lower-income sample consisted of participants enrolled in college or university and were more often exposed to the stress of facing a unique social setting.

Like motivations for enhancement and expansion, which are likely to occur due to the euphoria-inducing and hallucinogenic properties of cannabis [48,52], motivations for relaxation could be due to the anxiolytic properties of cannabis [39,40,41]. Notably, in pre-clinical studies, evidence shows that cannabidiol (CBD), the second major component of cannabis, was shown to be anxiolytic [58] by acting as a partial agonist of the 5HT_1A_ receptor [39,40,59], a positive allosteric modulator of GABA_A_ [60] and an enhancer of GABAnergic currents [61]. Moreover, in clinical studies, CBD had anxiolytic effects after four-week repeated CBD treatment of 18–19-year-old Japanese teens with SAD in a randomized, double-blind, placebo-controlled study [41].

In our study, relaxation was noted to be an equally important motivator for continuing and maintaining cannabis use in both groups. However, there were between-group distinctions in the reasons for wanting to relax. In the second stage of cannabis use, and like their respective controls, individuals with SAD reported using cannabis to relax from daily living. However, as they progressed to the maintenance stage, cannabis was seemingly being used to decompress from everyday stresses and overcome anxiety caused by social situations. In fact, individuals with SAD reported using cannabis to feel comfortable in social situations and even to replace the effects of antianxiety drugs prescribed to treat their SAD symptoms. The shift toward more coping with SAD symptoms may have indicated the progressive worsening of their SAD symptomatology and the lack of treatment options. However, using cannabis as a therapeutic tool could seemingly contribute to the development of CUD due to repeated exposure to cannabis and its addictive properties [30,49].

Although our qualitative research did not indicate that social motives were among the top four at each stage, our quantitative analysis showed that at the maintenance stage, using marijuana to be sociable, enjoy a party, or celebrate a special occasion with friends was more important to individuals with SAD. Moreover, social motivations were more likely to occur in participants with greater severity of SAD and those more likely to use cannabis to cope with their symptoms. Since individuals with SAD experienced fear and anxiety about social situations [1,2,3,4], using cannabis to deal with social settings was unsurprising.

Motivations for enhancement are associated with the psychomimetic properties of cannabis. Specifically, by acting through the cannabinoid receptors, δ-9-tetrahydrocannabinol (THC) stimulates neurons in the brain’s reward system (dorsal and ventral striatum) to release higher-than-normal dopamine levels, which are responsible for the euphoric mood in cannabis users [58,62,63]. The enhancement expectations appeared to be more motivating for the SAD group at the initial stages of cannabis use but equally motivating to continue and maintain cannabis use as in the control group. At the initial stage, using cannabis was expected to be a ‘fun activity’ to do with friends, which eventually grew into a desire to achieve an altered state and experience something new due to the psychoactive properties of marijuana. Interestingly, our study also demonstrated that individuals with SAD who were single or separated and those with lower income were more likely to use cannabis to get high and escape reality. Given that those with lower income would also have less access to psychological services that would help them cope with symptoms of SAD (as these services are not covered by Canadian universal healthcare) [1,2,3,4], it is not surprising that they would resort to cannabis as a coping strategy. Similarly, those living in isolation would also be more likely to use cannabis due to a lack of family support [1,2,3,4].

The motive for expansion is a self-directed goal designed to achieve positive outcomes [30,49]. The expansion was a theme used to code for behaviors resulting from the hallucinogenic properties of cannabis [30,49]. These behaviors included being more open to new experiences, expanding awareness, and being more creative and original [30,49]. Our research revealed that although expansion was not necessarily the initial motivation for cannabis use, it became an important reason for continuously using marijuana. Both groups of participants reported using marijuana to be more creative when writing a song, journaling, or re-evaluating a situation they have experienced. Moreover, within the SAD group, participants indicated they could cope with post-event ruminations more successfully because cannabis helped them process their thoughts more efficiently without feeling inadequate and anxious.

Although motivations for enhancement and expansion were not directly correlated to CUD, they may have led to pharmacological addiction as reward centers in the brain are associated with being motivated to seek euphoria and an altered state of mind [64,65]. Specifically, THC in cannabis induces dopamine release in the ventral tegmental area and nucleus accumbens by activating CB1 receptors [64,65]. Dopamine binds to dopamine receptors promoting pleasurable feelings associated with the rewarding behavior of drug use. Chronic exposure to cannabis decreases dopamine receptor density and metabolism in the reward system, reducing the ability of reward-seeking stimuli to activate the sensitivity of the reward system [64,65]. Thus, chronic cannabis users develop tolerance, as the same amounts of previously consumed cannabis can no longer activate the reward circuits. Hoping to achieve a ‘consistent high’ or the same degree of euphoria, the drug users begin to consume greater amounts of cannabis [64,65].

The prolonged suppression of the reward circuits by chronic cannabis use may lead to a sense of general depression, a state of anhedonia, and a lack of interest in previously enjoyable activities (otherwise termed pharmacological withdrawal), which may occur in the absence of drug use [64,65]. Consequently, drug use becomes more frequent, and it becomes the only activity that can activate the reward system strongly enough to evade the feelings of low mood associated with drug withdrawal [65].

### 4.2. Scientific Relevance

Given that the average age of initial cannabis use was determined to be 16 in the SAD group (high school level), in our study, we suggest that education programs demystifying cannabis use be introduced in grades 7 or 8 at the elementary school level. Such programs could educate early adolescents about the science behind cannabis and how it affects the developing brain. The focus could also be on familiarizing students with the public health policies and risks of using cannabis, including social science implications, peer pressure, decision making, harm reduction strategies, and how motivations for cannabis use change over time [66,67,68]. Such programs could also facilitate interaction with students who have previously struggled with CUD.

Promoting rewarding activities in vocational settings [68] may be beneficial, considering that motivations for enhancement and expansion were shown in our study to be significant at all three stages of cannabis use. For example, encouraging schoolchildren to participate in arts, crafts, drama, and acting clubs or getting them involved in playing music or singing may prevent the need to use cannabis to experience joy and pleasure [66,67,68]. Also, introducing the students to role models who abstain from cannabis may be preventive.

The motivations for social facilitation, coping, and relaxation could be mitigated specifically in socially anxious individuals by educating schoolchildren about the etiology and pathogenesis of SAD first, to identify those affected and, second, to encourage them to seek treatment. Bringing awareness to the negative impact of SAD is crucial, considering that a recent World Health Organization survey indicates that only 20% inflicted with SAD ever obtain treatment [5]. Information and knowledge dissemination about effective treatments for SAD is urgently needed in clinical settings.

### 4.3. Limitations

Several limitations to our research were noted and could be considered when conducting future studies. First, our recruitment strategy relied on individuals initiating contact by responding to our study advertisements. Since this was a non-referred sample (rather than a treatment-seeking clinical sample), participants’ personal experiences with SAD and cannabis use could not be confirmed by clinical examinations. Second, our data relied on participants’ recalls of events that led to motivations to initiate, continue, and maintain cannabis use rather than in-time reports of recent experiences. Recall of the events is more susceptible to memory bias than reporting incidents as they occur in time. Thereby, longitudinal studies with a group of participants with SAD and non-SAD controls could be conducted to track how opinions about cannabis and motivations for its use change over time from early adolescence into late adulthood. Such studies could also benefit from the convergent, mixed methods design with repeated measurement of both quantitative and qualitative research tools.

Third, due to budget and time limitations, our pilot study aimed to recruit only 26 participants in each group, allowing for us to account for the effect of age, sex, and current CUD when comparing the motivation for cannabis use between the two study groups. However, we could not conduct a more elaborate regression analysis between outcome measures and clinical and demographic variables for the SAD group due to sample size limitations. For this reason, future studies employing larger sample sizes may be needed to adequately assess the impact of a more significant number of demographic and clinical variables and investigate their moderating effect on the study results.

Fourth, the present study focused on current regular cannabis users. Future research could examine the motivations for cannabis use and the lack thereof between frequent, infrequent cannabis users, and nonusers with and without SAD. The purpose of such a study would be to elucidate the factors that would protect against cannabis use maladaptive behaviors. Nevertheless, our research provides important information about the distinctiveness of cannabis use patterns, problems, and motivations for cannabis use related to those diagnosed with moderate-to-severe SAD, enabling clinicians to better plan prevention and harm reduction efforts specific to this patient population.

## 5. Conclusions

In summary, individuals with SAD are more likely to exhibit problematic cannabis use and have a greater lifetime prevalence of CUD. Regarding motivations for cannabis use, curiosity is the primary motivation at the initial stages. At the same time, expectations for coping with SAD and ‘having a good time’ are more unique to the SAD group. Peer pressure or motivations for conformity appear to be more significant at the earlier stages and more relevant to adolescent social gatherings than solitary cannabis users with SAD. The top four motivations for continuing cannabis use in SAD are enhancement, coping, relaxation, and expansion, which persisted at the maintenance stage. At the maintenance stage, motivations for social facilitation and coping were specifically more pronounced in individuals with SAD, suggesting that better treatment for SAD and coping strategies are urgently needed to prevent those affected from falling back to cannabis as a primary coping tool. Promoting healthy coping strategies, such as engagement in enjoyable activities, could prevent the risk of those with social anxiety resorting to the intoxicating effects of marijuana. Psychoeducational and prevention efforts should inform how motivations for cannabis use change over time and how repeated cannabis use could eventually lead to problematic cannabis use and CUD.

## Figures and Tables

**Figure 1 brainsci-13-01698-f001:**
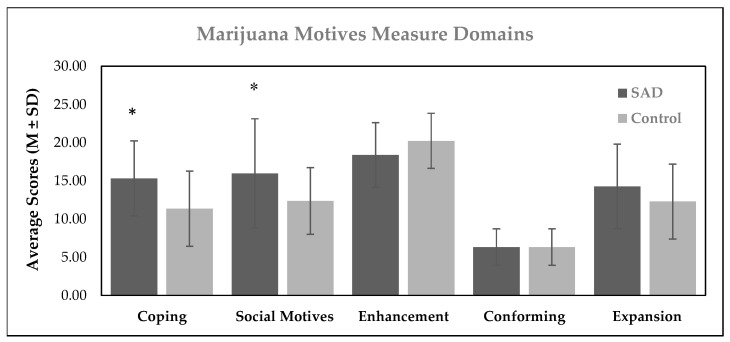
Cross-sectional comparison of Marijuana Motives Measure (MMM) domains. SAD = social anxiety disorder; M = mean; SD = standard deviation; * = *p* ≤ 0.05 compared to the control group. (n = 26 for the SAD and n = 26 for the control group).

**Figure 2 brainsci-13-01698-f002:**
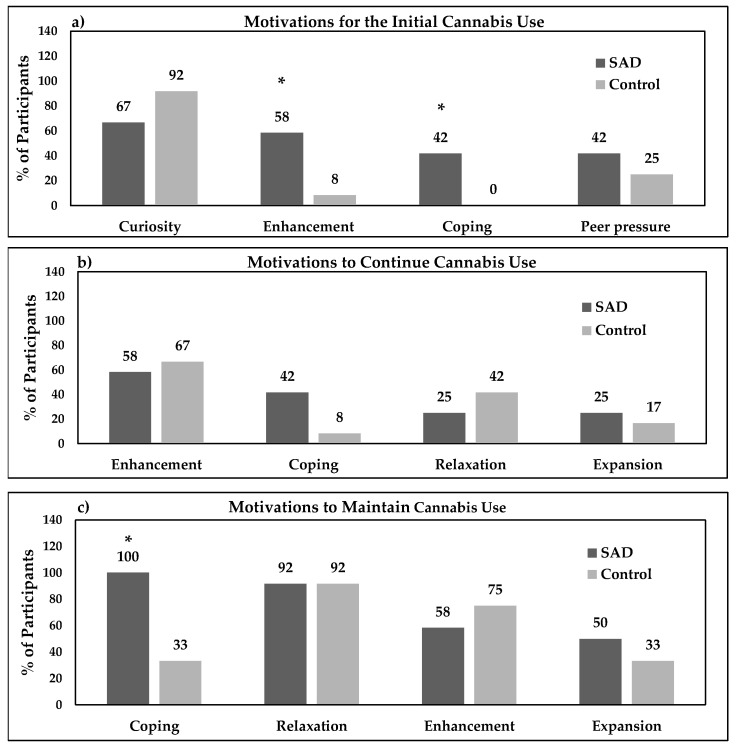
Cross-sectional comparison of motivations for cannabis use for n = 12 in SAD and n = 12 in the control group; SAD = social anxiety disorder; % = percentage; * *p* ≤ 0.05 compared to the control group.

**Table 1 brainsci-13-01698-t001:** Participant demographic and clinical characteristics.

	SAD (n = 26)	Controls (n = 26)	Statistic	Effect Size ^2^
** * Demographic Variables * **				
**Age (M ± SD)**	27.92 ± 7.34	27.35 ± 6.69	t = 0.30	0.08
**Sex (%F)**	53.8	50.0	χ^2^ = 0.08	0.04
**Highest Level of Education (%)**			χ^2^ = 1.95	0.19
Less than university	53.8	34.6		
University of higher	46.2	65.4		
**Years of Education (M ± SD)**	14.87 ± 1.94	15.62 ± 2.26	t = −1.28	0.36
**Race (%)**			χ^2^ = 2.58	0.22
East Asian	7.7	15.4		
South Asian	3.8	11.5		
Black African	7.7	11.5		
Caucasian	76.9	57.7		
Mixed race	3.8	3.8		
**Race (%)**			χ^2^ = 2.19	0.21
Other	23.1	42.3		
Caucasian	76.9	57.7		
**Race (%)**			χ^2^ = 0.75	0.12
Other	92.3	84.6		
East Asian	7.7	15.4		
**Occupational status (%)**			χ^2^ = 3.36	0.25
Unemployed	26.9	7.7		
Employed or Student	73.1	92.3		
**Income last year (%)**			χ^2^ = 3.50	0.27
<$50,000	60.0	33.3		
≥$50,000	40.0	66.7		
**Marital status (%)**			χ^2^ = 0.12	0.05
Single/Separated	76.9	80.8		
Married/Common Law	23.1	19.2		
** * Clinical variables * **				
**Age of onset of cannabis use (M ± SD)**	16.63 ± 3.15	20.12 ± 4.74	t = −3.12 *	0.87
**Age of heaviest cannabis use (M ± SD)**	23.05 ± 8.48	23.02 ± 4.50	t = 0.02	0.01
**Comorbidities (%)**				
Current CUD	57.7	46.2	χ^2^ = 0.70	0.12
Lifetime CUD	80.8	53.8	χ^2^ = 4.28 †	0.29
Past MDD	34.6			
Current GAD	26.9			
Other ^1^	30.8			
**LSAS total scores (M ± SD)**	92.65 ± 20.82	9.46 ± 6.52	t = 19.45 ‡	5.39
**CUPIT (M ± SD)**	36.54 ± 12.35	26.31 ± 9.80	t = 3.31 †	0.92
**Amount of cannabis use/week (g)**	7.57 ± 9.37	2.00 ± 1.81	t = 2.92 †	0.82
**Weekly frequency of cannabis use**	5.06 ± 2.19	3.96 ± 2.28	t = 1.77	0.49

* *p* < 0.05, † *p* < 0.01, ‡ *p* < 0.001. SAD = social anxiety disorder; CUD = Cannabis Use Disorder; MDD = major depressive disorder; GAD = generalized anxiety disorder; LSAS = Liebowitz Social Anxiety Scale; CUPIT = Cannabis Use Problems Identification Test; F = female; M = mean; SD = standard deviation; g = grams; % = percentage; χ^2^ = chi-square test; t = independent *t*-test. ^1^ Other co-morbidity: past alcohol use disorder, other specified eating disorder, current nicotine use disorder, past nicotine use disorder, current persistent depressive disorder, past persistent depressive disorder past bulimia nervosa, current major depressive disorder, post-traumatic stress disorder not otherwise specified, and other specified depressive disorder. ^2^ Cohen’s d was included for parametric and Phi Coefficient for non-parametric data.

**Table 2 brainsci-13-01698-t002:** Marijuana Motives Measure (MMM) domains in the maintenance stage (n = 52).

	Dependent Variables														
	MMM Coping			MMM Social			MMM Enhancement			MMM Conformity			MMM Expansion		
Predictor Variables	B (SE)	β	η^2^	B (SE)	β	η^2^	B (SE)	β	η^2^	B (SE)	β	η^2^	B (SE)	β	η^2^
**^1^** **SAD status**	3.43 (1.13)	0.33 †	0.16	3.38 (1.61)	0.28 *	0.04	−0.21 (1.01)	−0.25	0.08	0.14 (0.59)	0.03	0.0	1.96 (1.48)	0.19	0.04
**^1^** **Sex**	−0.86 (1.13)	−0.08	0.01	−0.24 (1.61)	−0.02	0.0	−1.32 (1.01)	−0.17	0.04	0.21 (0.58)	0.05	0.0	0.45 (1.48)	0.04	0.0
**Age**	−0.12 (0.08)	−0.15	0.04	−0.16 (0.12)	−0.18	0.04	−0.13 (0.07)	−0.23	0.07	−0.04 (0.04)	−0.13	0.02	−0.11 (0.11)	−0.14	0.02
**^1^** **Current CUD**	5.45 (1.15)	0.52 ‡	0.33	2.98 (1.63)	0.25	0.07	2.57 (1.02)	0.32 *	0.12	0.95 (0.60)	0.23	0.05	0.73 (1.51)	0.07	0.01

* *p* < 0.05, † *p* < 0.01, ‡ *p* < 0.001. n = number of participants; MMM = Marijuana Motives Measure; CUD = cannabis use disorder; SAD = social anxiety disorder; B = unstandardized coefficient beta; SE = standard error; β = standardized coefficient beta; η^2^ = partial eta square; asterisk indicates *t*-test significant *p*-values. ^1^ dummy coding for sex (0 = male, 1 = female); current CUD (0 = no, 1 = yes); SAD status (0 = no; 1 = yes).

**Table 3 brainsci-13-01698-t003:** Bivariate correlation between demographic and clinical measures for participants with social anxiety disorder (SAD).

(n = 26)	Age	Sex ^1^	Current CUD ^1^	Lifetime MDD ^1^	Current GAD ^1^	Educ. ^1^	Race (W) ^1^	Race (EA) ^1^	Income ^1^	Occ. Status ^1^	Marital Status ^1^
**Total LSAS scores**	−0.16	−0.06	0.28	0.19	0.02	−0.14	0.18	−0.03	−0.29	0.26	−0.25
**Weekly cannabis use frequency**	0.14	0.21	0.60 †	0.06	0.28	−0.39 *	0.55 †	0.34	−0.28	0.07	−0.24
**The weekly amount of cannabis use (g)**	0.13	0.23	0.42 *	−0.02	0.46 *	−0.59 †	−0.24	0.18	−0.35	0.09	−0.14
**CUPIT**	0.13	0.16	0.65 †	−0.02	0.41 *	−0.43 *	−0.27	0.20	−0.55 †	0.13	−0.20
**MMM Total**	−0.55 †	0.20	0.51 †	0.14	0.11	0.07	−0.13	0.15	−0.34	−0.11	−0.46 *
**MMM Coping**	−0.51 †	0.11	0.60 †	0.17	0.27	0.14	0.06	0.11	−0.43 *	0.02	−0.32
**MMM Social**	−0.38	0.08	0.32	−0.03	0.24	−0.11	−0.11	0.06	−0.12	−0.29	−0.32
**MMM Enhancement**	−0.32	−0.06	0.38	0.24	0.07	0.21	0.00	0.13	−0.43 *	−0.20	−0.54 †
**MMM Conformity**	−0.27	0.22	0.38	−0.19	−0.02	0.12	0.05	0.16	0.05	−0.23	−0.24
**MMM Expansion**	−0.33	0.33	0.10	0.19	−0.26	0.01	−0.31	0.09	−0.19	0.29	−0.16

* *p* < 0.05; † *p* < 0.01; values represent Pearson’s correlation coefficients r. n = number of participants; MMM = Marijuana Motives Measure; LSAS = Liebowitz Social Anxiety Scale; CUPIT = Cannabis Use Problem Identification Test; CUD = Cannabis Use Disorder; MDD = major depressive disorder; GAD = generalized anxiety disorder; Educ. = highest levels of education; W = White race; EA = East Asian race; Occ. Status = occupational status; g = grams. ^1^ dummy coding for sex (0 = male, 1 = female); current CUD, lifetime MDD, and current GAD (0 = no, 1 = yes); for highest level of education (0 = less than university, 1 = university education or higher); race (W), (0 = other, 1 = White); race (EA), (0 = other, 1 = East Asian); income (0 = <$50,000, 1 = ≥$50,000), occupational status (0 = unemployed, 1 = employer or student); marital status (0 = single or separated, 1 = married or common law).

**Table 4 brainsci-13-01698-t004:** Simple bivariate correlation between clinical variables for participants with social anxiety disorder (SAD).

(n = 26)	1	2	3	4	5	6	7	8	9
**1. Total LSAS scores**									
**2. Weekly cannabis use frequency**	0.16								
**3. The weekly amount of cannabis use (g)**	0.28	0.56 †							
**4. CUPIT**	0.37	0.70 †	0.78 †						
**5. MMM Total**	0.43 *	0.12	0.26	0.23					
**6. MMM Coping**	0.49 *	0.22	0.21	0.36	0.80 †				
**7. MMM Social**	0.50 †	0.31	0.18	0.17	0.82 †	0.59 †			
**8. MMM Enhancement**	0.27	−0.17	0.05	0.16	0.69 †	0.53 †	0.39 *		
**9. MMM Conformity**	0.13	0.12	0.07	0.05	0.41 *	0.33	0.36	0.17	
**10. MMM Expansion**	−0.06	−0.16	0.26	−0.01	0.50 *	0.16	0.15	0.25	−0.1

* *p* < 0.05, † *p* < 0.01; values represent Pearson’s correlation coefficients r. n = number of participants; LSAS = Liebowitz Social Anxiety Scale; CUPIT = Cannabis Use Problems Identification Test; MMM = Marijuana Motives Measure; g = grams.

## Data Availability

The data are not publicly available due to their sensitive nature, and our ethical approval prevents us from sharing data beyond named collaborators. Further inquiries can be directed to the corresponding author B.L.F.

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
