# Peer review of "Motivations for Cannabis Use in Individuals with Social Anxiety Disorder (SAD)"

_brainsci, 2023, doi:10.3390/brainsci13121698_

Round 1

Reviewer 1 Report

Comments and Suggestions for Authors

In the manuscript entitled "Motivations for cannabis use in individuals with 2 Social Anxiety Disorder (SAD)" the authors aimed at investigate the motivations behind cannabis use in people with SAD, using a mixed methods approach. 
Although the relatively small sample, the paper is well structured and gives a good insight into this interesting topic. 
I would only suggest to expand the Discussion section by further investigating the different components of cannabis (CBD and THC) and their quite opposite effects on several mental health conditions, ranging from mood to anxiety, and even psychomimetic properties.
I suggest to refere to: Bartoli F, Riboldi I, Bachi B, et al. Efficacy of Cannabidiol for Δ-9-Tetrahydrocannabinol-Induced Psychotic Symptoms, Schizophrenia, and Cannabis Use Disorders: A Narrative Review. J Clin Med. 2021;10(6):1303. Published 2021 Mar 22. 

Author Response

Dear Reviewer 1,

Thank you so much for reviewing our article titled: “Motivations for Cannabis Use in Individuals with Social Anxiety Disorder (SAD)”.  We have drafted our responses to your suggestions and requests below:

  1. In the manuscript entitled "Motivations for cannabis use in individuals with 2 Social Anxiety Disorder (SAD)" the authors aimed at investigate the motivations behind cannabis use in people with SAD, using a mixed methods approach.

Although the relatively small sample, the paper is well structured and gives a good insight into this interesting topic.

I would only suggest to expand the Discussion section by further investigating the different components of cannabis (CBD and THC) and their quite opposite effects on several mental health conditions, ranging from mood to anxiety, and even psychomimetic properties.

I suggest to refere to: Bartoli F, Riboldi I, Bachi B, et al. Efficacy of Cannabidiol for Δ-9-Tetrahydrocannabinol-Induced Psychotic Symptoms, Schizophrenia, and Cannabis Use Disorders: A Narrative Review. J Clin Med. 2021;10(6):1303. Published 2021 Mar 22.

Thank you for your suggestion.  The focus of our discussion is on motivations for cannabis use in SAD, and as such we feel that a broader discussion of components of cannabis, CBD, and THC and their effects on mental health disorders is outside the scope of this article.  However, we have added the following information on CBD and THC, linking their pharmacological effects to the specific motivations for cannabis use:

  • Lines 637 – 643: “Notably, in the pre-clinical studies, the evidence shows that cannabidiol (CBD), the second major component of cannabis, was shown to be anxiolytic [58] by acting as a partial agonist of the 5HT1A receptor [39,40.59], a positive allosteric modulator of GABAA [60] and an enhancer of GABAnergic currents [61]. Moreover, in clinical studies, CBD had anxiolytic effects after four-week repeated CBD treatment of 18-19-year-old Japanese teens with SAD in a randomized, double-blind, placebo-controlled study [41].”
  • Lines 665 – 669: “Motivations for enhancement are associated with the psychomimetic properties of cannabis. Specifically, by acting through the cannabinoid receptors, δ-9-tetrahydrocannabinol (THC), stimulates neurons in the brain's reward system (dorsal and ventral striatum) to release higher-than-normal dopamine levels, which is responsible for the euphoric mood in cannabis users [58,62,63].”
  • Lines 696 – 710: “Specifically, THC in cannabis induces dopamine release in the ventral tegmental area and nucleus accumbens by activating CB1 receptors [64.65]. Dopamine binds to dopa-mine receptors promoting pleasurable feelings associated with the rewarding behavior of drug use. Chronic exposure to cannabis decreases dopamine receptor density and metabolism in the reward system, reducing the ability of reward-seeking stimuli to activate the sensitivity of the reward system [64.65].  Thus, chronic cannabis users develop tolerance, as the same amounts of previously consumed cannabis can no longer activate the reward circuits.  Hoping to achieve a ‘consistent high’ or the same degree of euphoria, the drug users begin to consume greater amounts of cannabis [64.65]

The prolonged suppression of the reward circuits by chronic cannabis use may lead to a sense of general depression, a state of anhedonia, and a lack of interest in previously enjoyable activities (otherwise termed pharmacological withdrawal), which may occur in the absence of drug use [64,65].  Consequently, drug use becomes more frequent, and it becomes the only activity that can activate the reward system strongly enough to evade the feelings of low mood associated with drug withdrawal [65].   

The paper by Bartoli et al., 2021 was cited and is listed as one of our references.

Thank you again for reviewing our manuscript.  Please do not hesitate to let us know should you need further clarification regarding our responses.

Sincere regards,

Sonja Elsaid on behalf of all authors

Reviewer 2 Report

Comments and Suggestions for Authors

This study aimed to understand the differences in cannabis use motivation between individuals with and without Social Anxiety Disorder (SAD) at the maintenance stage of cannabis use. The study is unique in its focus on adults with moderate-to-severe SAD and examines the trajectory of cannabis use motivations. The authors have conducted the study well. They have written the manuscript well. However, the reviewer has some questions and suggestions-

  • 1. The ages of onset for SAD range from 11 to 27 years, as mentioned. This is a broad range and could benefit from clarification or a reference to studies that specifically define this range.
  • 2. The paper states that individuals with SAD are "almost seven times more likely to develop CUD" but does not specify the comparison group. Is this in comparison to individuals without any mental health disorder, or compared to those with other anxiety disorders? Clarification would be helpful.
  • 3. The paper mentions the study aims to understand "motivations to start, continue, and maintain cannabis use" but doesn't clearly distinguish these different phases. It would be beneficial to define what is meant by each (e.g., "start" vs. "continue" vs. "maintain").
    • 4. The introduction seems to jump between different topics, such as the prevalence of SAD, its comorbidity with CUD, and then back to the prevalence and impact of SAD. A more linear structure might enhance readability.
    • 5. Some statements might be too broad or require more specific evidence. For example, the claim that SAD is "hard-to-treat" could be nuanced by discussing what makes it challenging compared to other psychiatric conditions.

Author Response

Dear Reviewer 2,

Thank you so much for reviewing our article titled: “Motivations for Cannabis Use in Individuals with Social Anxiety Disorder (SAD)”.  We have drafted our responses to your suggestions and requests below:

This study aimed to understand the differences in cannabis use motivation between individuals with and without Social Anxiety Disorder (SAD) at the maintenance stage of cannabis use. The study is unique in its focus on adults with moderate-to-severe SAD and examines the trajectory of cannabis use motivations. The authors have conducted the study well. They have written the manuscript well. However, the reviewer has some questions and suggestions-

  1. The ages of onset for SAD range from 11 to 27 years, as mentioned. This is a broad range and could benefit from clarification or a reference to studies that specifically define this range.

To make it more clear, we have now altered the statement: The average age of onset ranges from 11-27 years [4], indicating that this debilitating mental health condition occurs early in life, leading to significant functional and psychosocial impairments. ”  (lines 39 – 40) to the following:

“The risk period for onset of SAD ranges between mid-to-late adolescence to early 40s, worldwide [4]. Individuals with SAD experience functional and psychosocial impairments in several aspects of their daily life [1-4].” (lines 39 – 41)

Also, please note that the information related to prevalence and age of onset of SAD is taken from a research article: Stein, D. J.; Lim, C. C. W.;  Roest, A. M.;  de Jonge, P.;  Aguilar-Gaxiola, S.;  Al-Hamzawi, A.;  Alonso, J.;  Benjet, C.;  Bromet, E. J.;  Bruffaerts, R.;  de Girolamo, G.;  Florescu, S.;  Gureje, O.;  Haro, J. M.;  Harris, M. G.;  He, Y.;  Hinkov, H.;  Horiguchi, I.;  Hu, C.;  Karam, A.;  Karam, E. G.;  Lee, S.;  Lepine, J.-P.;  Navarro-Mateu, F.;  Pennell, B.-E.;  Piazza, M.;  Posada-Villa, J.;  ten Have, M.;  Torres, Y.;  Viana, M. C.;  Wojtyniak, B.;  Xavier, M.;  Kessler, R. C.;  Scott, K. M.; Collaborators, W. H. O. W. M. H. S., The cross-national epidemiology of social anxiety disorder: Data from the World Mental Health Survey Initiative. BMC Medicine 2017, 15 (1), 143.

  1. The paper states that individuals with SAD are "almost seven times more likely to develop CUD" but does not specify the comparison group. Is this in comparison to individuals without any mental health disorder, or compared to those with other anxiety disorders? Clarification would be helpful.

To increase the clarity, we have now altered the sentence: “Individuals with SAD are seven times more likely to develop CUD after eliminating the effects of other comorbid anxiety diagnoses, depression and other substance use and personality disorders” (lines 52 – 55) to the following:

Compared to the general population, individuals with SAD are almost seven times more likely to develop CUD after eliminating the effects of other comorbid anxiety diagnoses, depression, and other substance use and personality disorders [7,9-17]” (lines 56 – 57)

  1. The paper mentions the study aims to understand "motivations to start, continue, and maintain cannabis use" but doesn't clearly distinguish these different phases. It would be beneficial to define what is meant by each (e.g., "start" vs. "continue" vs. "maintain").

We have now altered the sentence (lines 69 – 72) “Also, while most studies were correlational in nature and examined current motivations for cannabis use in undergraduate students with social anxiety [10,11,14,18,19,22], motivations to start, continue, and maintain cannabis use in individuals with SAD remains unclear” to read:

“Also, while most studies were correlational in nature and examined current motivations for cannabis use in undergraduate students with social anxiety [10,11,14,18,19,22], motivations to start (for the first time), continue immediately after the initial use, and maintain cannabis use up to the assessment time point remains unclear in individuals with SAD”. (lines 73 – 76)

  1. The introduction seems to jump between different topics, such as the prevalence of SAD, its comorbidity with CUD, and then back to the prevalence and impact of SAD. A more linear structure might enhance readability.

Our Introduction section follows the ‘funnel approach’, discussing the broader topics first and ending with our specific project objectives and hypotheses, as follows:

  1. The broader demographic topics related to SAD first prevalence, age of onset, impairment leading to socioeconomic burden (lines 37 – 54)
  2. Next, we discussed SAD being the risk factor for developing CUD (lines 55 – 59)
  3. Then, are the findings from previous studies investigating motivations for cannabis use in individuals with SAD (lines 60 – 69)
  4. Next is our rationale for conducting this study, and the advantages of using the mixed methods approach (lines 70 – 84)
  5. Then, we discussed our study objectives (lines 85 – 87)
  6. Lastly, we stated our hypotheses (lines 87 – 92)
  7. Some statements might be too broad or require more specific evidence. For example, the claim that SAD is "hard-to-treat" could be nuanced by discussing what makes it challenging compared to other psychiatric conditions.
  • To eliminate the confusion, we have replaced the word ‘hard-to-treat’ with ‘debilitating’ in our abstract (line 19).
  • Furthermore, in the Introductions section (lines 43 – 52), we have added information describing difficulties in treating SAD. The added sentence reads: According to The World Health (WHO) World Mental Health Survey, 1 in 5 individuals with SAD seek treatment and of those the probability of receiving helpful treatment is 92%, but only after seeing up to seven different healthcare professionals [5].

Thank you again for reviewing our manuscript.  Please do not hesitate to let us know should you need further clarification regarding our responses.

Sincere regards,

Sonja Elsaid on behalf of all authors
